# Current Status of Multimodal Therapy for Oligometastatic Disease, Induced Oligometastatic Disease, and Oligo-Progressive Disease in *EGFR*-Mutated Non-Small-Cell Lung Cancer

**DOI:** 10.3390/cancers17132202

**Published:** 2025-06-30

**Authors:** Taichi Miyawaki, Hirotsugu Kenmotsu, Ryo Ko, Masaki Oshima, Takehito Shukuya, Naoto Shikama, Kazuhisa Takahashi

**Affiliations:** 1Department of Respiratory Medicine, Graduate School of Medicine, Juntendo University, Tokyo 113-8421, Japan; tshukuya@juntendo.ac.jp (T.S.); kztakaha@juntendo.ac.jp (K.T.); 2Division of Thoracic Oncology, Shizuoka Cancer Center, Shizuoka 411-8777, Japan; h.kenmotsu@scchr.jp (H.K.); r.ko@scchr.jp (R.K.); 3Department of Radiation Oncology, Graduate School of Medicine, Juntendo University, Tokyo 113-8421, Japan; msoosima@juntendo.ac.jp (M.O.); n-shikama@juntendo.ac.jp (N.S.)

**Keywords:** *EGFR* mutation, non-small-cell lung cancer, oligometastatic disease, oligo-residual disease, oligo-progressive disease, EGFR-TKI, local ablative therapy

## Abstract

The treatment landscape for *EGFR*-mutated non-small-cell lung cancer (NSCLC) has significantly evolved with the introduction of EGFR tyrosine kinase inhibitors (EGFR-TKIs). However, acquired resistance to EGFR-TKIs remains inevitable, limiting long-term survival. Recent evidence suggests that local ablative therapy (LAT), such as stereotactic radiotherapy or surgery, may improve outcomes in patients with a limited number of metastatic lesions—collectively referred to as oligometastatic states. These include synchronous oligometastatic disease (present at diagnosis), oligo-residual disease (persistent lesions after initial EGFR-TKI therapy), and oligo-progressive disease (limited progressive disease during ongoing EGFR-TKI treatment). Integrating LAT with systemic therapy may delay resistance, prolong disease control, and potentially improve survival. However, optimal timing and patient selection for LAT remain uncertain, and standardized criteria for defining these “oligo” states are still lacking. This review summarizes the current clinical evidence, ongoing trials, and future perspectives for the role of LAT in the multidisciplinary management of *EGFR*-mutated NSCLC across different oligometastatic settings.

## 1. Introduction

Epidermal growth factor receptor (EGFR)-tyrosine kinase inhibitors (TKIs) have shown clinical activity for patients with *EGFR*-mutated non-small-cell lung cancer (NSCLC). Multiple clinical trials have shown that EGFR-TKIs provide better efficacy than conventional platinum-based chemotherapy in patients with *EGFR*-mutated NSCLC [1,2,3]. In addition, a pivotal phase III study showed that treatment with osimertinib, as a third-generation EGFR-TKI, has provided superior efficacy compared with treatment with first-generation EGFR-TKIs, including erlotinib and gefitinib [4,5]. Recent pivotal clinical trials have led to an evolution in the standard of care for patients with *EGFR*-mutated NSCLC. The FLAURA2 and MARIPOSA studies demonstrated that novel combination therapies confer significant clinical benefits in patients with EGFR mutated non-small-cell lung cancer (NSCLC). Specifically, the combination of amivantamab and lazertinib in the MARIPOSA trial, as well as the combination of platinum-based chemotherapy and osimertinib in the FLAURA2 trial, both showed improved progression-free survival and favorable efficacy profiles compared with the current standard of care, osimertinib monotherapy [6,7]. However, the development of resistance to EGFR-TKIs remains almost inevitable and continues to pose a major challenge to achieving prolonged survival in patients [8]. Although multiple resistance mechanisms to third-generation EGFR-TKIs—such as C797S mutation, MET amplification, and BRAF mutations—have been identified, effective strategies to fully overcome these challenges remain elusive [9]. “Oligometastatic disease” was first conceptualized by Hellman and Weichselbaum in 1995 as an intermediate state between localized and widespread systemic disease, defined by a limited number and sites of metastases. Initially proposed in the context of stage IV breast cancer, this paradigm has since been recognized across multiple tumor types, suggesting a distinct biological state potentially amenable to curative-intent local therapies [10]. Initially, oligometastatic disease described a specific condition in which there were few metastatic lesions at the time of diagnosis. However, the concept has since evolved to encompass broader clinical scenarios, including oligo-persistence (oligo-residual disease after systemic therapy) and oligo-progressive disease (Oligo-PD; limited progression with otherwise effective systemic treatment) [11]. These broader definitions underscore a deeper understanding of the biological heterogeneity inherent in metastatic dissemination and have important therapeutic implications, particularly with the integration of local ablative therapies into multimodal treatment strategies [11].

Local ablative therapy (LAT) has shown promising high local control of involved lesions and potential survival benefits for patients with oligometastatic NSCLC [12,13,14]. Furthermore, in patients with oligo-residual disease—where individuals with *EGFR*-mutated NSCLC transition to an oligometastatic state following a period of EGFR-TKI therapy—there is emerging evidence suggesting that the addition of local ablative therapy (LAT) to target residual lesions may confer further clinical benefit [15]. Additionally, Oligo-PD refers to the progression of a limited number of existing or new lesions during EGFR-TKI treatment. In such cases, additional local ablative therapy (LAT) for the progressive lesions, combined with the continuation of EGFR-TKI therapy, has been shown to be potentially effective and may offer clinical benefit [16].

This review highlights evolving multidisciplinary treatment strategies for patients with *EGFR*-mutated NSCLC. It focuses on current and emerging therapeutic options tailored to specific clinical scenarios, including oligometastatic disease at diagnosis, oligo-residual disease emerging during EGFR-TKI treatment, and oligo-progressive disease after the development of resistance. By exploring these distinct disease states, we aim to provide practical insights into how local and systemic therapies can be integrated to optimize patient outcomes.

## 2. Oligometastatic Disease in *EGFR*-Mutated NSCLC

### 2.1. Concept of Oligometastatic Disease

Hellman and Weichselbaum originally proposed oligometastatic disease as an intermediate state between local and advanced breast cancer [10]. Oligometastatic disease is increasingly recognized as a distinct clinical state in which cancer exhibits limited metastatic potential, resulting in a slower progression and fewer lesions. In particular, synchronous oligometastatic disease refers to cases where both the primary tumor and a small number of metastases are present simultaneously, offering a unique therapeutic window in which all visible disease sites may be effectively targeted with LAT.

In patients with oligometastatic NSCLC, progressive disease (PD) after first-line systemic therapy has been shown to be substantially limited to the involved sites of disease (Figure 1). Retrospective studies examining the patterns of progression of oligometastatic NSCLC patients have demonstrated that 70–90% of the patients develop PD limited to previously involved disease sites after first-line systemic therapy [17,18]. Furthermore, almost half of the patients with *EGFR*-mutated NSCLC have PD limited to previous localized disease after first-line treatment with EGFR-TKIs [19,20,21]. Synchronous oligometastatic NSCLC accounts for a variable prevalence of approximately 20–30% of advanced NSCLC and represents a not-rare population [18,22].

Previous studies and guidelines have defined synchronous oligometastatic NSCLC differently, with the definitions including 1–3 metastases or 1–5 metastases [18,23,24,25]. To date, no standard definition of synchronous oligometastatic NSCLC has been identified. Patients with synchronous oligometastatic NSCLC have been frequently defined as those with 1–3 metastases or 1–5 metastases in recent phase III trials [26,27,28,29].

Furthermore, in a survey by the European Organization for Research and Treatment of Cancer (EORTC), nearly half of the respondents indicated 1–3 metastases as the criteria for synchronous oligometastatic NSCLC [24]. However, the European Multidisciplinary Consensus Group proposed the criteria for synchronous oligometastatic disease as 1–5 metastases and 1–3 metastatic organs [25]. Furthermore, a single-institution retrospective study suggested that the presence of 1–3 metastases may serve as a reasonable criterion for defining synchronous oligometastatic NSCLC, based on observed patterns of progression following first-line systemic therapy [18]. Furthermore, in a survey with 444 physicians in the EORTC, about half of the respondents answered that the standard number of metastases for synchronous oligometastatic disease of NSCLC would be 1–3 metastases [24].

Additionally, patients with malignant pleural effusion, pericardial effusion, pleural dissemination, meningeal dissemination, peritoneal dissemination, ascites, and cancerous lymphangitis should be excluded from synchronous oligometastatic disease.

### 2.2. Clinical Trials for Oligometastatic Disease

Several clinical trials have suggested that LAT may offer survival benefits in patients with oligometastatic NSCLC, regardless of driver gene mutation status [12,30,31,32]. In a randomized phase II trial enrolling patients with oligometastatic NSCLC, the incorporation of LAT into standard systemic treatment resulted in significantly improved clinical outcomes compared with systemic therapy alone. The median PFS was 14.2 months versus 4.4 months (*p* = 0.022), and the median OS was 41.2 months versus 17.0 months (*p* = 0.017) [13]. Another randomized phase II trial in patients with oligometastatic NSCLC demonstrated significantly improved efficacy with the addition of LAT (*p* = 0.01), reporting a median progression-free survival (PFS) of 9.7 months in the LAT group compared to 3.5 months in the maintenance therapy group following initial systemic treatment [12].

The was a randomized phase II trial evaluating the role of stereotactic ablative radiotherapy (SABR) in patients with a controlled primary tumor and 1–5 metastatic lesions. The study demonstrated that the addition of SABR to standard palliative care significantly improved both OS and PFS in patients with oligometastatic solid cancer. Specifically, median OS was 41.0 months in the SABR group versus 28.0 months in the control group (*p* = 0.006), and median PFS was 11.6 months compared to 5.4 months (*p* = 0.001) [14].

In recent years, the first prospective study to evaluate the efficacy of a multidimensional approach combining immune checkpoint inhibitor (ICI) therapy and local ablative therapy (LAT) in patients with driver mutation-negative or unknown NSCLC has been reported. This is a single-arm, phase II study that evaluated the efficacy of pembrolizumab following radical local therapy (LAT) in 45 patients with metastatic NSCLC. The primary endpoint of progression-free survival from the start of LAT (PFS-L) was 19.1 months, significantly exceeding the previous benchmark of 6.6 months (*p* = 0.005) [33].

Multidisciplinary treatment development has also progressed in *EGFR*-mutated NSCLC with oligometastatic disease. In a recent phase III trial involving patients with *EGFR*-mutated oligometastatic NSCLC, the addition of LAT to first-generation EGFR-TKIs (LAT + TKI arm) significantly improved both PFS and OS compared to EGFR-TKIs alone (TKI arm), with median PFS of 20.2 vs. 12.5 months (*p* < 0.001) and median OS of 25.5 vs. 17.4 months (*p* < 0.001), respectively [30].

A recent phase III trial demonstrated that the addition of thoracic radiotherapy (TRT) to EGFR-TKI therapy significantly improved survival in patients with *EGFR*-mutated NSCLC and oligo-organ metastases. Among 136 patients with ≤5 metastatic lesions in ≤2 organs, median PFS was 13.8 vs. 10.1 months (hazard ratio [HR] 0.56; *p* = 0.006) and median OS was 31.2 vs. 24.5 months (HR 0.59; *p* = 0.030) in the TRT and control arms, respectively [34].

Notably, these studies utilized first-generation EGFR-TKIs, rather than third-generation agents, which have since become the standard therapeutic approach in this clinical setting [4]. Furthermore, the frequency of synchronous oligometastatic disease in patients with *EGFR*-mutated NSCLC constitutes a rare population (6%) [15]. Therefore, the clinical benefit of incorporating LAT in *EGFR*-mutated NSCLC patients with synchronous oligometastatic disease remains insufficiently established.

Furthermore, it remains unclear whether the combination of LAT with third-generation EGFR-TKIs provides optimal benefits, necessitating clinical trials investigating the efficacy of a combination of third-generation EGFR-TKIs and LAT. A single-arm phase II study evaluating the efficacy of osimertinib plus LAT in *EGFR*-mutated NSCLC patients with synchronous oligometastatic disease is currently ongoing (NCT04908956). In addition, a randomized phase II trial comparing lazertinib, as a third-generation EGFR-TKI, plus LAT to lazertinib alone for patients with synchronous oligometastatic *EGFR*-mutated NSCLC has been ongoing (NCT05167851). The results of these ongoing clinical trials for patients with *EGFR*-mutated NSCLC and synchronous oligometastatic disease could provide the basis for new therapeutic approaches in the future. In addition, a randomized phase II trial comparing molecular-targeted therapy plus LAT to residual lesions versus molecular-targeted therapy alone is ongoing for patients with *EGFR*-mutated/*ALK*-rearranged oligometastatic NSCLC (NCT05277844) (Table 1).

## 3. Oligo-Residual Disease in *EGFR*-Mutated NSCLC

### 3.1. Concept of Oligo-Persistent Disease/Oligo-Residual Disease

In the ESTRO/EORTC consensus on the characterization and classification of oligometastatic disease, the concepts of oligo-residual and oligo-persistent disease were introduced to reflect evolving clinical scenarios in which patients, initially diagnosed with polymetastatic cancer, achieve a limited number of residual or persistent metastases following systemic therapy—highlighting the dynamic nature of metastatic progression and the need to redefine oligometastasis not only by the number of lesions but also by disease biology and treatment response [11]. Although most patients with *EGFR*-mutated NSCLC respond to EGFR-TKIs, nearly all patients develop EGFR-TKI resistance. Unfortunately, few effective salvage systemic therapy options are available after the development of resistance to EGFR-TKIs [35,36]. Complete responses remain rare even with EGFR-TKIs, and almost all patients have a certain burden of residual disease, even with the best response. The addition of LAT to residual disease during the best response to EGFR-TKIs has been hypothesized to eliminate resistant clones and potentially lead to a better response duration (Figure 2) [37].

In metastatic NSCLC or colorectal cancer, oligo-residual disease (also known as oligo-persistent disease) is a newly proposed disease setting, and several retrospective studies have shown that LAT at all sites of residual disease shows favorable efficacy in patients with oligo-residual disease [31,38,39,40]. Oligo-residual disease represents a state of conversion from multiple initial metastases to oligometastatic disease induced by effective systemic therapy.

The results of a previous study showed that the majority of patients with oligo-residual disease had PD limited to residual sites after EGFR-TKI treatment, providing a rationale for LAT of all disease sites in patients with *EGFR*-mutated NSCLC and oligo-residual disease. Furthermore, only 6% of the patients had synchronous oligometastatic disease before treatment, which increased to 32% after treatment. These findings suggest that EGFR-TKI treatment for 3 months increases the number of patients eligible for LAT by more than fivefold [15]. In addition, osimertinib treatment was found to be an independent predictor of oligo-residual disease and PD limited to the residual sites. Therefore, osimertinib has replaced the standard of care for *EGFR*-mutated NSCLC patients, which suggests that oligo-residual diseases are more significant.

Recent studies have defined oligo-residual disease by the presence of 1–4 residual lesions, including the primary site, at 3–6 months after the initiation of EGFR-TKI therapy. Residual disease was defined by the presence of detectable lesions on imaging, 3–6 months after EGFR-TKI therapy. The definition of oligo-residual disease varies from trial to trial, and no definitive criteria have yet been identified.

### 3.2. Clinical Trials for Oligo-Residual Disease

The first large retrospective study of oligo-residual disease divided patients into three groups according to the category of LAT to residual sites: the all-LAT group (patients who received LAT to all residual lesions, including primary sites, lymph nodes, and metastases), the part-LAT group (patients who received LAT to primary sites or other metastatic sites), and the non-LAT group (patients who received no prior LAT anywhere). The median OS in the all-LAT, part-LAT, and non-LAT groups was 40.9, 34.1, and 30.8 months, respectively (*p* < 0.001). This retrospective study revealed that consolidative LAT may improve the survival outcomes of *EGFR*-mutated NSCLC patients who had oligo-residual disease during first-line EGFR-TKI therapy [31].

However, few prospective studies have evaluated whether the addition of LAT to residual disease provides survival benefits. A recent single-arm phase II trial explored the efficacy of adding LAT to the treatment of residual disease in patients with *EGFR*-mutated NSCLC who achieved oligo-residual disease after 3 months of EGFR-TKI treatment. Among the 16 patients included in the analysis, the median PFS and OS for EGFR-TKI were 15.2 months and 43.3 months, respectively. Furthermore, patients included in the analysis had significantly better PFS than those who failed screening, suggesting that additional LAT was associated with a reduced risk of PD (HR: 0.41, *p* = 0.0097) [38].

A recent single-arm phase II trial evaluated consolidative stereotactic radiotherapy in 61 patients with metastatic *EGFR*-mutated NSCLC who had oligo-residual disease after first-line third-generation EGFR-TKI treatment. The median PFS was 29.9 months, exceeding the predefined threshold. A propensity score-matched analysis showed significantly longer PFS in the SRT group compared to EGFR-TKI alone (HR 0.46; *p* = 0.002) [41].

In a recent randomized phase II trial (NCT03595644), 62 patients with stage IV *EGFR*-mutated NSCLC (exon 19 deletion or L858R) who responded to first-line EGFR-TKI therapy and had 1–5 metastatic residual lesions were enrolled. In the TKI + SBRT group, patients received SBRT to the residual lesions along with continued EGFR-TKI therapy, while in the control group, patients received EGFR-TKI therapy alone. The TKI + SBRT group demonstrated significantly prolonged PFS (17.6 vs. 9.0 months, HR 0.52, *p* = 0.016) and OS (33.6 vs. 23.2 months, HR 0.53, *p* = 0.026). These findings suggest that adding SBRT to residual lesions in patients with oligo-residual disease may delay the development of acquired resistance and improve outcomes in *EGFR*-mutated NSCLC [42].

Another randomized phase II study of osimertinib with or without local consolidation therapy for *EGFR*-mutated NSCLC patients with oligo-residual disease (NCT03410043) is currently ongoing. A randomized phase II study of consolidative radiotherapy for residual lesions during osimertinib monotherapy (ORIHALCON trial/WJOG13920L, JRCTs041220115) evaluated patients who received osimertinib monotherapy for at least 90 days but not more than 120 days at the time of enrollment after the initiation of first-line osimertinib monotherapy. Patients with residual disease at the time of enrollment, based on the following definitions, with a total of three or fewer residual lesions. The results of these prospective randomized trials highlight the significance of additional LAT for patients with NSCLC with oligo-residual disease (Table 2).

## 4. Current Status of Oligo-PD

### 4.1. Concept of Oligo-PD

Oligo-progression (Oligo-PD) might be a relatively new concept that has emerged with the development of more effective systemic therapies, indicating the progression of a few lesions in patients whose disease has been extensively controlled by systemic therapy [11,43]. A recent consensus of the American Radium Society (ARS) recommends that the criteria for Oligo-PD be restricted to 1–3 sites of progression [44]. Furthermore, previous studies showed that the addition of LAT to the lesions of progression may provide additional survival benefit in patients with Oligo-PD during PD-1/PD-L1 inhibitor monotherapy [45,46]. Focal/regional progression (with overall disease control at other sites) can occur due to tumor heterogeneity, with the selective pressure of EGFR-TKIs causing the outgrowth of subclones with pre-existing or acquired alterations conveying a fitness advantage. The resistant subclones may be spatially confined to distinct anatomical sites because of factors such as local microenvironments and the timing of branch points in clonal evolution with respect to site seeding. This pattern of localized drug resistance or pharmacokinetic failure raises the possibility that local therapies can be used to address resistant subclones or sanctuary sites and prolong the clinical benefits of ongoing TKI therapy.

Previous studies have shown that LAT may eliminate resistant subclones in a few progressive lesions and prevent local symptoms and further complications from tumor proliferation [32]. On the basis of these principles, the addition of LAT may be beneficial for patients with Oligo-PD during pembrolizumab monotherapy or PD-1/PD-L1 inhibitor therapy plus chemotherapy.

Oligo-PD is generally assumed to be a consequence of the development of isolated resistant subclones that occur only at a few metastatic sites and emerge when more effective systemic therapies are available. In *EGFR*-mutated NSCLC, Oligo-PD has been proposed to be histologically heterogeneous, with coexisting tumor cells that are resistant and sensitive to EGFR-TKIs [47,48].

Although approximately half of patients with *EGFR*-mutated NSCLC have been suggested to develop PD during treatment with EGFR-TKIs, and this has been exclusively attributed to existing lesions, it remains unclear whether the pattern of disease progression during EGFR-TKI treatment varies according to the number of metastatic lesions (Figure 3). The results of the FLURA trial have shown that the proportion of patients with PD limited to residual disease was higher in the osimertinib arm, at 81%, compared to 63% in the first-generation EGFR-TKI arm [5]. In addition, a recent nationwide cohort study in Switzerland showed that 77% of 147 patients who received osimertinib in the first-line setting had Oligo-PD [49].

As recommended by the most recent ESMO and NCCN guidelines, local ablative therapy (LAT) should be considered for patients with oligo-progressive disease (OPD) [50,51]. This strategy enables targeted control of limited sites of progression, allowing continued benefit from the ongoing systemic therapy and offering a potential survival advantage without compromising the patient’s overall treatment plan.

### 4.2. Definition of Oligo-PD

The frequency of Oligo-PD during EGFR-TKI treatment has been suggested to be 15–47% [44]. In clinical trials of Oligo-PD in previously treated NSCLC, the criteria for the number of metastases varied from trial to trial. However, no common criteria have yet been established. The consensus definition of Oligo-PD generally assumes that patients exclusively have progressive disease (up to two or five lesions) with slower tumor biological features. The American Radium Society (ARS) recently proposed a criterion of 1–3 progressive lesions for Oligo-PD in NSCLC.

In advanced NSCLC patients with *EGFR* mutations, LAT for Oligo-PD after EGFR-TKI therapy may eradicate the lesions, including resistant clones [52,53,54]. A metastasis-directed treatment approach may prolong the duration of targeted therapy and delay the transition to the next systemic therapy as long as possible, thereby achieving the maximum longevity of TKI efficacy and contributing to prolonged survival. Advancements in radiotherapy have contributed to new treatment strategies for *EGFR*-mutated NSCLC with Oligo-PD owing to the availability of stereotactic radiotherapy (SRT), an advanced radiotherapy modality with high local tumor control rates.

### 4.3. Current Status of Clinical Trials for Oligo-PD

The CURB trial represents the first randomized clinical study worldwide to focus exclusively on patients with Oligo-PD. This randomized phase II trial enrolled patients with breast cancer or NSCLC presenting with 1–5 progressive lesions (Oligo-PD) and compared the efficacy of LAT to all progressive lesions (LAT arm) versus the physician’s choice of systemic chemotherapy (chemotherapy arm). The primary endpoint, PFS, was significantly prolonged in the LAT arm compared to the chemotherapy arm, with median PFS of 7.2 months versus 3.2 months, respectively (HR 0.53, *p* = 0.0035). Notably, in the subgroup analysis of patients with NSCLC, the median PFS was 10.0 months in the LAT arm and 2.2 months in the chemotherapy arm, again demonstrating a significant benefit in favor of LAT (*p* = 0.001). In contrast, OS did not differ significantly between the treatment arms in the overall population or within the NSCLC and breast cancer cohorts. This lack of OS difference has been attributed to several factors, including the relatively short follow-up period and substantial heterogeneity in baseline assessment methods, treatment lines, therapeutic modalities, and tumor burden [55].

Other previous studies on Oligo-PD after PD-1/PD-L1 inhibitor treatment in patients with NSCLC have also not identified the benefits of LAT for patients with Oligo-PD [56,57]. However, these studies included heterogeneous populations in terms of pretreatment systemic therapy, including molecular-targeted therapy as well as immune checkpoint inhibitors, and the number of treatment sequences. The frequency of Oligo-PD in *EGFR*-mutated NSCLC after treatment with first- and second-generation EGFR-TKIs has previously been shown to be 15–45%, while the frequency of Oligo-PD after osimertinib has been reported to be 73–75%.

The frequency of Oligo-PD in patients with advanced NSLC after treatment with ICI has been described to be approximately 20–30%. Furthermore, the frequency of Oligo-PD after a therapeutic response to ICI was 56%. Oligo-PD may be more frequent with more effective treatment.

### 4.4. Ongoing Trials for Oligo-PD

Information regarding the ongoing clinical trials for patients with *EGFR*-mutated NSCLC with Oligo-PD is shown in Table 3. A single-arm phase II trial is ongoing to evaluate the efficacy of furmonertinib, a third-generation EGFR-TKI, combined with radiotherapy for *EGFR*-mutated NSCLC patients with Oligo-PD (NCT04970693). Additionally, two single-arm phase II trials (NCT02759835, NCT04216121) are ongoing to evaluate the efficacy of osimertinib combined with LAT in patients with *EGFR*-mutated Oligo-PD NSCLC. Furthermore, the results of these prospective trials will help establish a multidisciplinary treatment regimen for *EGFR*-mutated NSCLC with Oligo-PD.

The HALT trial (NCT03256981) is an ongoing international, multicenter, randomized phase II/III study evaluating the efficacy of SBRT in patients with stage IV NSCLC harboring actionable mutations who develop Oligo-PD (≤5 lesions). Patients are randomized 2:1 to receive SBRT for all progressive lesions, followed by continued TKI or TKI alone [58].

## 5. Local Ablative Therapy

The mainstay of LAT for oligometastatic NSCLC has been recently shifted to RT, in particular SBRT. The role of RT in the treatment of metastatic NSCLC has historically been limited to palliative purposes [59]. Conventional low-dose RT has been shown to be effective in palliating symptoms associated with metastases at different sites of the body. Conventional irradiation has provided symptomatic relief, yet irradiated tumors often re-expand after some time and have not contributed to long-term local control or prolonged survival [60].

Stereotactic body radiation therapy (SBRT) is a new technique of radiation therapy that delivers a precise dose to the target tumor in several doses equal to or greater than conventional radical radiotherapy, is relatively safe, and provides a high degree of local control of lesions within the field of radiation [61]. Furthermore, the cytological mechanism of the therapeutic effect of SBRT has been shown to result not only in direct cytotoxicity, as well as microvascular damage and endothelial apoptosis, leading to the death of perfused tissue [62]. SBRT has been the predominant type of LAT in the majority of clinical trials for the treatment of oligometastatic, oligo-residual, and oligo-progressive diseases.

### 5.1. Primary Site and Lung Metastases

Surgery has traditionally been the mainstay of LAT for chest lesions such as primary tumors and lung metastases in patients with oligometastatic NSCLC. Surgery for primary tumors or lung metastases in patients with oligometastatic NSCLC has been shown to contribute to long-term survival since the concept of oligometastatic disease was introduced. Currently, in highly selected cohorts of patients with oligometastatic NSCLC, pulmonary resection still has the potential to contribute to long-term survival and local control.

Dramatic advances in radiation techniques have led to the advent of high-dose radiotherapy, including SBRT, which now dominates LAT for primary and lung metastases in patients with oligometastatic NSCLC [60]. A recent observational study evaluating the efficacy of SBRT for primary site or lung metastases in patients with oligometastatic NSCLC has demonstrated 2-year local control rates of 88.9% and no grade 4 pulmonary toxicity [63]. Furthermore, a retrospective study evaluating the efficacy of SBRT for lung metastases in 140 patients with oligometastatic solid tumors showed 2-year local control rates of 85.1% with only 1.7% grade 2 pulmonary toxicity [64]. Due to the favorable local control and safety of SBRT to lung lesions, SBRT has been the mainstay of treatment in most of the previous and ongoing clinical trials in oligometastatic NSCLC. In cases involving central tumors, T3 or higher primary lesions, or mediastinal lymph node involvement, conventional three-dimensional radiotherapy with doses ranging from 45/15 Fr, 60 Gy/30 fr, or 66 Gy/33 Fr has been employed, resulting in favorable local control.

In contrast, stereotactic body radiotherapy (SBRT) is typically applied for peripheral or T1–T2 tumors. According to Collen et al., the local control rate for thoracic lesions was favorable, reaching approximately 89%.

A recent randomized phase III clinical trial (Northern Radiation Oncology Group of China-002) evaluated the efficacy and safety of first-line thoracic radiotherapy (TRT) combined with EGFR-tyrosine kinase inhibitors (TKIs) in patients with oligo-organ metastatic non-small-cell lung cancer (NSCLC) harboring *EGFR*-mutations, compared with TKI monotherapy.

The combination of TKI plus TRT significantly improved survival, with a median progression-free survival (PFS) of 17.1 months versus 10.6 months in the TKI alone group (HR 0.57; *p* = 0.004) and a median overall survival (OS) of 34.4 months versus 26.2 months (HR 0.62; *p* = 0.029).

In terms of safety, the TKI plus TRT group experienced higher rates of serious adverse events, including radiation esophagitis (6.8%), radiation pneumonitis (5.1%), and leukopenia or neutropenia (3.4%), all of which were significantly more common than in the TKI monotherapy group [34].

### 5.2. Brain Metastases

Stereotactic radiosurgery (SRS) has been widely performed for a few brain metastases, replacing the previously widely performed whole-brain radiation therapy (WBRT), due to fewer neurocognitive side effects [65]. In a phase III trial comparing stereotactic radiotherapy alone with stereotactic radiotherapy plus WBRT for patients with four or fewer metastatic brain tumors, median overall survival was 8 months in the stereotactic radiotherapy alone group and 7.5 months in the stereotactic radiotherapy plus WBRT group, with no significant difference [66]. Furthermore, a phase III trial comparing the efficacy of WBRT plus SRS versus WBRT alone in patients with solid tumors and 1–3 brain metastases demonstrated a significant survival benefit in the WBRT + SRS group (median OS: 6.5 vs. 4.9 months, *p* = 0.0393) [67].

Based on the results of several studies, SRS for brain metastases has been performed in most clinical trials for oligometastatic NSCLC, resulting in favorable local control [12,68,69].

### 5.3. Adrenal Gland Metastases

The advent of SBRT in radiation oncology has provided a promising option for the local treatment of adrenal gland metastases [70]. SBRT has been widely applied as a local treatment for adrenal gland metastases in NSCLC with oligometastatic disease or Oligo-PD, with favorable local control achieved [55,71,72,73]. In addition, side effects of SBRT for adrenal metastases have been shown to be rare, minimal, and well managed [74,75]. As an alternative to SBRT, high-dose conventional radiation has been applied for adrenal gland metastases, such as 60 Gy/30 Fr or 45 Gy/15 Fr, which has resulted in a favorable local control [12,13,76].

### 5.4. Bone Metastases

Bone metastases are often associated with pain, as well as fractures, spinal cord compression, and hypercalcemia, which can significantly reduce the physical and functional quality of life. The most effective treatment for palliation of pain due to bone metastases from solid tumors is radiotherapy, including single and multiple fractions [77,78]. SBRT is a new option for radiotherapy of bone metastases and has been shown to improve local control and pain relief in patients with spinal metastases [79].

Most clinical trials in patients with oligometastatic solid tumors have applied SBRT as spinal cord irradiation and have shown high local control rates of 75–95% [80,81,82]. Recent phase II/III trials have shown that the risk of local failure and re-irradiation for spinal cord metastases has been lower with SBRT than with conventional radiation therapy [83].

A previous study of 106 solid tumors with non-spine bone metastases revealed a favorable 2-year local control rate of approximately 85% for stereotactic radiotherapy for non-spine bone metastases [84]. In addition, a recent randomized phase II trial comparing conventional radiotherapy and SBRT for bone metastases in non-spinal bone metastases showed 24-month local control rates of 100% in the SBRT group, which was significantly better than the conventional radiotherapy group (*p* = 0.02) [85].

### 5.5. Liver Metastases

Several retrospective studies have evaluated the efficacy of SBRT for liver metastases and have shown favorable 3-year local control of 70–90% [86,87,88]. In a recent large registry study, SBRT for liver metastases showed favorable local control, 87% at 1 year and 68% at 3 years. In addition, the incidence of grade 3 or higher toxicity was 3.9% in the study, indicating that the safety of SBRT for liver metastases seems to be ensured [89].

## 6. Summary

This review discusses the multidisciplinary treatment strategy for patients with *EGFR*-mutated NSCLC, focusing on the appropriate timing of LAT in combination with EGFR-TKIs for those with synchronous oligometastatic disease, oligo-residual disease, and Oligo-PD.

The findings indicate that LAT can be considered not only before EGFR-TKI treatment but also during EGFR-TKI treatment and even after EGFR-TKI failure, depending on the oligo status of the patient (synchronous oligometastatic disease, oligo-residual disease, or Oligo-PD).

A recent phase III trial showed better survival with additional LAT, indicating that LAT could be considered for the treatment of synchronous oligometastatic disease in patients with *EGFR*-mutated NSCLC [30,34]. Synchronous oligometastatic disease may be rare, accounting for only 6% of patients with advanced *EGFR*-mutated NSCLC [15], while oligo-residual disease accounts for approximately 30% of *EGFR*-mutated NSCLC patients. Even if patients have poly-metastatic disease at the start of the initial therapy and if they have oligo-residual disease after 3–6 months of EGFR-TKI treatment, LAT for residual disease could be considered.

Oligometastatic disease or oligo-residual disease represents a state prior to resistance to EGFR-TKI treatment, whereas Oligo-PD represents a state after resistance to EGFR-TKI treatment and may be a different clinical entity. Synchronous oligometastatic disease, oligo-residual disease, or poly-metastatic disease can potentially develop into Oligo-PD after treatment with EGFR-TKIs or LAT. To avoid the potential loss of additional LAT, Oligo-PD should be reviewed in patients with progressive disease during EGFR-TKI treatment, regardless of previous treatment. Meanwhile, the criteria for synchronous oligometastatic, oligo-residual, and Oligo-PD in patients with *EGFR*-mutated NSCLC have not been adequately established, and it is necessary to establish globally common criteria to stimulate therapeutic development. Furthermore, it remains unclear whether the addition of LAT to EGFR-TKI treatment confers a survival benefit for both oligometastatic disease, oligo-residual disease, and Oligo-PD. Several ongoing clinical trials should address these issues and establish LAT as a standard treatment for all oligo-diseases in the foreseeable future.

## 7. Conclusions

LAT has emerged as a potentially valuable option for managing synchronous oligometastatic, oligo-residual, and oligo-progressive disease in *EGFR*-mutated NSCLC. Further clarification of disease definitions and results from ongoing clinical trials will be essential to determine the appropriate role of LAT in these distinct clinical states.

## Figures and Tables

**Figure 1 cancers-17-02202-f001:**
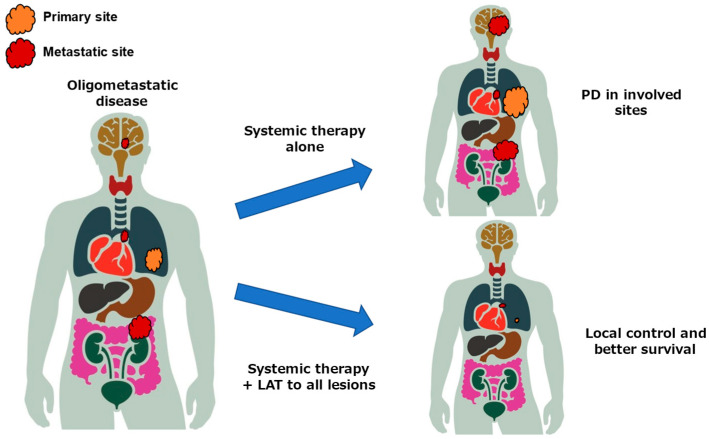
Concept of oligometastatic disease and failure patterns of systemic therapy. *EGFR,* epidermal growth factor receptor; NSCLC, non-small-cell lung cancer; TKI, tyrosine kinase inhibitor; LAT, local ablation therapy.

**Figure 2 cancers-17-02202-f002:**
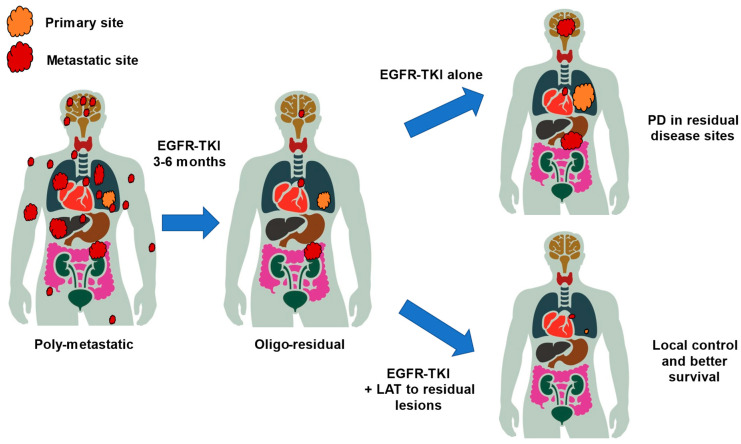
Concept of oligo-residual disease and patterns of PD after treatment with EGFR-TKI. EGFR, epidermal growth factor receptor; PD, progressive disease; TKI, tyrosine kinase inhibitor; LAT, local ablative therapy.

**Figure 3 cancers-17-02202-f003:**
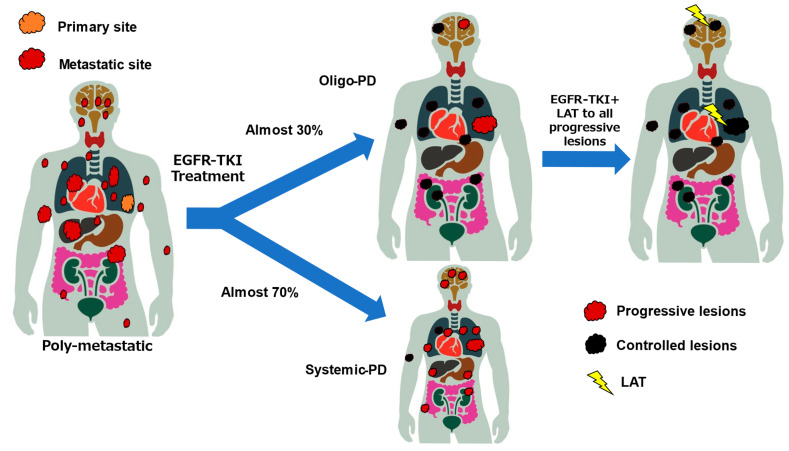
Concept of oligo-progressive disease, patterns of PD after treatment with EGFR-TKI, and frequency of Oligo-PD. *EGFR*, epidermal growth factor receptor; PD, progressive disease; TKI, tyrosine kinase inhibitor; LAT, local ablative therapy.

**Table 1 cancers-17-02202-t001:** Trials for *EGFR*-mutated NSCLC with oligometastatic disease.

Trial	Status	Criteria for Metastatic Lesions	Study Design	Primary Outcome	Treatment
NCT02893332 (Sindas study)	Completed	1–5	Phase III	PFS	1st-generation EGFR-TKI with SBRT to all lesions vs. 1st-generation EGFR-TKI alone
NCT04908956	Recruiting	1–5	Single-arm phase II	Safety PFS	Osimertinib with SBRT to all lesions
NCT05167851	Not yet recruiting	1–5	Randomized phase II	PFS	Lazertinib with SBRT to all lesions vs. Lazertinib alone
NCT05277844 (TARGET-01)	Recruiting	1–5	Randomized phase II	PFS	EGFR-TKI (or ALK-TKI) with SBRT to all lesions vs. EGFR-TKI (or ALK-TKI) alone *

NSCLC, non-small-cell lung cancer; PFS, progression-free survival; EGFR, epidermal growth factor receptor; TKI, tyrosine kinase inhibitor; SBRT, stereotactic body radiotherapy; *ALK*, anaplastic lymphoma kinase. * The study included *EGFR*-mutated patients as well as *ALK* translocated patients.

**Table 2 cancers-17-02202-t002:** Trials for *EGFR-mutated* NSCLC with oligo-residual disease.

Trial	Status	Criteria for Residual	Length of Induction EGFR-TKI Treatment	Study Design	Primary Outcome	Treatment
NCT01941654 (ATOM study)	Completed	1–5	12–14 weeks	Single-arm phase II	PFS	SBRT to all residual disease following induction of treatment with 1st-generation EGFR-TKI alone
NCT03410043 (NORTHSTAR Study)	Not recruiting	N.A.	6–12 weeks	Randomized phase II	PFS	Surgery or RT to all residual disease following induction of treatment with Osimertinib vs. Osimertinib alone
jRCTs041220115 (ORIHALCON trial/WJOG13920L)	Recruiting	1–3	90–120 days	Randomized phase II	PFS	SBRT to all residual disease following induction of treatment with Osimertinib vs. Osimertinib alone

NSCLC, non-small-cell lung cancer; PFS, progression-free survival; EGFR, epidermal growth factor receptor; TKI, tyrosine kinase inhibitor; SBRT, stereotactic body radiotherapy; RT, radiotherapy; N.A., not available.

**Table 3 cancers-17-02202-t003:** Clinical trial for patients with *EGFR*-mutated NSCLC with oligo-progressive disease.

Trial	Criteria for Progressive Lesions	Study Design	Primary Outcome	Treatment
NCT04970693	3–5	Non-randomized phase II	PFS	Furmonertinib with RT to all progressive lesions
NCT02759835	N.A	Non-randomized phase II	PFS	Osimertinib with LAT to all progressive lesions
NCT04216121 (LAT-FLOSI)	1–3	Observational study	PFS2	Osimertinib with LAT to all progressive lesions
NCT03256981 (HALT)	1–3	Randomized phase II/III study	PFS	SBRT for all progressive lesions, followed by continued TKI vs. continued TKI I

NSCLC, non-small-cell lung cancer; PFS, progression-free survival; *EGFR*, epidermal growth factor receptor; TKI, tyrosine kinase inhibitor; RT, radiotherapy; LAT, local ablative therapy; SBRT, stereotactic body radiotherapy.

## Data Availability

No new data were created or analyzed in this study. Data sharing is not applicable to this article.

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
