# Peer review of "Current Status of Multimodal Therapy for Oligometastatic Disease, Induced Oligometastatic Disease, and Oligo-Progressive Disease in EGFR-Mutated Non-Small-Cell Lung Cancer"

_cancers, 2025, doi:10.3390/cancers17132202_

Round 1
Reviewer 1 Report
Comments and Suggestions for Authors
Thank you for your nice review which is very complete. I had great pleasure to readit and I have only a few comments:
should you not mention the type of metastatic disease which are not suitable for this approach (pleural effusions;..)
another point you have just approach, is the local treatment for the primary disease. You mention surgery and SBRT but there is also the question of classical chest radiotherapy.
Could you comment on this point
Thank you
Author Response
Thank you very much for your thoughtful and constructive comments on our review. We are truly grateful for your kind words and for taking the time to read our manuscript so thoroughly.
In response to your first point, we agree that it is important to clarify the types of metastatic disease that are not suitable for a multimodal treatment approach. As you suggested, we have added the following statement to the manuscript:
“Additionally, patients with malignant pleural effusion, pericardial effusion, pleural dissemination, meningeal dissemination, peritoneal dissemination, ascites, and cancerous lymphangitis should be excluded from synchronous oligometastatic disease.”
Regarding your second comment on local treatment for the primary tumor, we appreciate your suggestion to elaborate on this important aspect. In addition to surgery and SBRT, we have now included a description of conventional thoracic radiotherapy in the revised manuscript. Specifically, we added the following:
“In cases involving central tumors, T3 or higher primary lesions, or mediastinal lymph node involvement, conventional three-dimensional radiotherapy with doses ranging from 45/15Fr, 60Gy/30fr or 66 Gy/33Fr have been employed, resulting in favorable local control.”
We sincerely thank you again for your valuable feedback, which has helped us improve the quality and completeness of our review.
Reviewer 2 Report
Comments and Suggestions for Authors
The article provides a well-structured and insightful review of current and emerging treatment strategies for EGFR-mutated non-small cell lung cancer (NSCLC), particularly focusing on the roles of local ablative therapy (LAT) in the context of oligometastatic, oligo-residual, and oligo-progressive disease states. It effectively outlines the clinical challenges posed by resistance to EGFR-tyrosine kinase inhibitors (TKIs), including third-generation agents like osimertinib, and underscores the relevance of combining systemic therapies with localized interventions to enhance patient outcomes.
The manuscript is well written, presents current literature clearly, and provides a comprehensive overview that will be of interest to clinicians and researchers in the field of oncology. With some minor corrections this article would be a valuable addition to the existing body of literature.
Recommendation: I recommend this article for publication following minor revisions.
Specific recommendations:
Abstract:
- Please correct „oligometastatic disease“: who have oligometastatic disease, oligo-residual disease during treatment with EGFR-TKIs, and oligo-progressive disease following resistance to EGFR-TKIs.
Oligometastatic disease can bi „de novo“, synchronous, metachronous, oligo-induced, oligo-progresive, oligo-residual… I believe you were referring to „de novo“ or synchronous oligometastatic disease?
- Introduction:
- Local ablative therapy (LAT) has shown promising high regional control of involved lesions and potential survival benefits for patients with oligometastatic NSCLC.
Please correct regional control to local control.
- In patients with Oligo-PD, additional LAT for progressive lesions and continued EGFR-TKI therapy may be more effective than conventional salvage chemotherapy.
Please explain how the CURB trial is the reference of LAT vs. 2nd line chemotherapy. As CURB was SoC systemic treatment vs. SoC sist. treatment + SBRT. The cohort was mixed breast cancer and NSCLC and only 14% of lung cancer patients had driver mutation (3 patients in SOC and 5 patients in SBRT arm).
- Oligometastatic disease in EGFR-mutated NSCLC
- In patients with oligometastatic NSCLC, progressive disease (PD) after first-line chemotherapy has been shown to be substantially limited to the involved sites of disease (Figure 1).
Chemotherapy? In Figure 1 there is TKI. Please correct.
- In recent years, the first prospective study to evaluate the efficacy of a multidimen-sional approach combining immune checkpoint inhibitor (ICI) therapy and local ablative therapy (LAT) in patients with driver mutation-negative or unknown NSCLC has been reported. This is a single-arm, phase II study that evaluated the efficacy of pembrolizumab following radical local therapy (LAT) in 45 patients with metastatic NSCLC. The primary endpoint of progression-free survival from the start of LAT (PFS-L) was 19.1 months, sig-nificantly exceeding the previous benchmark of 6.6 months (P = 0.005) 37.
The reference 37 is wrong. Please correct.
5.4. Bone metastases
Conventionally, low-dose irradiation has been predominantly applied for the prevention or palliation of skeletal-related adverse events (SREs) associated with bone metastases.
The statement is incorrect. Prophylactic bone irradiation for prevention of SRE is not conventional treatment, since evidence in the literature is scarse. Please correct.
References:
Following trials are missing in the text, and as references. I suggest them to be added.
https://pubmed.ncbi.nlm.nih.gov/37105304/
https://pubmed.ncbi.nlm.nih.gov/39398493/
https://www.jto.org/article/S1556-0864(22)01202-3/pdf
Author Response
Recommendation: I recommend this article for publication following minor revisions.
Specific recommendations:
Abstract:
- Please correct „oligometastatic disease“: who have oligometastatic disease, oligo-residual disease during treatment with EGFR-TKIs, and oligo-progressive disease following resistance to EGFR-TKIs.
Oligometastatic disease can bi „de novo“, synchronous, metachronous, oligo-induced, oligo-progresive, oligo-residual… I believe you were referring to „de novo“ or synchronous oligometastatic disease?
Thank you very much for your valuable and insightful comment. In accordance with your suggestion, we have revised the terminology to “synchronous oligometastatic disease.”
- Introduction:
- Local ablative therapy (LAT) has shown promising high regional control of involved lesions and potential survival benefits for patients with oligometastatic NSCLC.
Please correct regional control to local control.
I sincerely appreciate your valuable comment. We have revised the relevant section to read “local control.”
- In patients with Oligo-PD, additional LAT for progressive lesions and continued EGFR-TKI therapy may be more effective than conventional salvage chemotherapy.
Please explain how the CURB trial is the reference of LAT vs. 2nd line chemotherapy. As CURB was SoC systemic treatment vs. SoC sist. treatment + SBRT. The cohort was mixed breast cancer and NSCLC and only 14% of lung cancer patients had driver mutation (3 patients in SOC and 5 patients in SBRT arm).
We are sincerely grateful for your valuable and constructive feedback. As you rightly pointed out, the CURB trial was not an appropriate reference in this context. We have accordingly replaced the citation with:
Monica F. Chen et al. Outcomes After Radiation for Oligoprogressive Disease Sites in Patients With EGFR-Mutant Lung Cancer Treated With Osimertinib. JCO Precis Oncol 9, e2500047 (2025).
Furthermore, we have revised the relevant description as follows:
“Additionally, Oligo-PD refers to the progression of a limited number of existing or new lesions during EGFR-TKI treatment. In such cases, additional local ablative therapy (LAT) for the progressive lesions, combined with the continuation of EGFR-TKI therapy, has been shown to be potentially effective and may offer clinical benefit.”
- Oligometastatic disease in EGFR-mutated NSCLC
- In patients with oligometastatic NSCLC, progressive disease (PD) after first-line chemotherapy has been shown to be substantially limited to the involved sites of disease (Figure 1).
Chemotherapy? In Figure 1 there is TKI. Please correct.
⇒We greatly appreciate your insightful comment. In response, we have revised the manuscript to consistently use the term “systemic therapy” in place of separate references to TKI and chemotherapy. Figure 1 has also been updated to reflect this change.
- In recent years, the first prospective study to evaluate the efficacy of a multidimen-sional approach combining immune checkpoint inhibitor (ICI) therapy and local ablative therapy (LAT) in patients with driver mutation-negative or unknown NSCLC has been reported. This is a single-arm, phase II study that evaluated the efficacy of pembrolizumab following radical local therapy (LAT) in 45 patients with metastatic NSCLC. The primary endpoint of progression-free survival from the start of LAT (PFS-L) was 19.1 months, sig-nificantly exceeding the previous benchmark of 6.6 months (P = 0.005) 37.
The reference 37 is wrong. Please correct.
⇒We apologize for the oversight in our previous reference. The citation has been corrected to the appropriate source:
"Pembrolizumab After Completion of Locally Ablative Therapy for Oligometastatic Non-Small Cell Lung Cancer: A Phase 2 Trial." J. M. Bauml, R. Mick, C. Ciunci, C. Aggarwal, C. Davis, T. Evans, et al. JAMA Oncology, 2019; 5(9): 1283–1290.
5.4. Bone metastases
Conventionally, low-dose irradiation has been predominantly applied for the prevention or palliation of skeletal-related adverse events (SREs) associated with bone metastases.
The statement is incorrect. Prophylactic bone irradiation for prevention of SRE is not conventional treatment, since evidence in the literature is scarse. Please correct.
⇒We are sincerely grateful for your thoughtful and constructive feedback. After careful consideration of your comment, we have decided to remove the relevant section from the manuscript to ensure accuracy and clarity.
References:
Following trials are missing in the text, and as references. I suggest them to be added.
https://pubmed.ncbi.nlm.nih.gov/37105304/
⇒We are deeply grateful for your important suggestion. In response, we have added the following description to section 3.3. Clinical trials for oligo-residual disease and cited the corresponding reference:
“In a recent randomized phase II trial (NCT03595644), 62 patients with stage IV EGFR-mutated NSCLC (exon 19 deletion or L858R) who responded to first-line EGFR-TKI therapy and had 1–5 metastatic residual lesions were enrolled. In the TKI+SBRT group, patients received SBRT to the residual lesions along with continued EGFR-TKI therapy, while in the control group, patients received EGFR-TKI therapy alone. The TKI+SBRT group demonstrated significantly prolonged PFS (17.6 vs. 9.0 months, HR 0.52, P = 0.016) and OS (33.6 vs. 23.2 months, HR 0.53, P = 0.026). These findings suggest that adding SBRT to residual lesions in patients with oligo-residual disease may delay the development of acquired resistance and improve outcomes in EGFR-mutated NSCLC.⁴⁸”
https://pubmed.ncbi.nlm.nih.gov/39398493/
⇒Thank you very much for introducing this important reference. We have cited this study and added the following description to section 3.3. Clinical trials for oligo-residual disease:
“A recent single-arm phase II trial evaluated consolidative stereotactic radiotherapy in 61 patients with metastatic EGFR-mutated NSCLC who had oligo-residual disease after first-line third-generation EGFR-TKI treatment. The median PFS was 29.9 months, exceeding the predefined threshold. A propensity score-matched analysis showed significantly longer PFS in the SRT group compared to EGFR-TKI alone (HR 0.46; P = 0.002)⁴⁷.”
https://www.jto.org/article/S1556-0864(22)01202-3/pdf
⇒Thank you very much for introducing this important clinical trial. In response, we have added the following description to section 4.4. Ongoing trials for Oligo-PD:
“The HALT trial (NCT03256981) is an ongoing international, multicenter, randomized phase II/III study evaluating the efficacy of SBRT in patients with stage IV NSCLC harboring actionable mutations who develop Oligo-PD (≤5 lesions). Patients are randomized 2:1 to receive SBRT for all progressive lesions followed by continued TKI or TKI alone.”
Additionally, we have included the HALT trial (NCT03256981) in Table 3.
Reviewer 3 Report
Comments and Suggestions for Authors
This is a comprehensive, up-to-date and very well written review on several related and clinically important subjects: OMD, Oligopersistence, and OPD. The material is rich and structured, there is hardly something to add.
Optionally, the authors could consider to also mention the CURB study (NCT03808662), which was the first randomized study to demonstrate benefit from LAT in oligoprogressive NSCLC including tumors with oncogenic drivers. Of note, no benefit was noted for breast cancer, which shows that lung cancer maybe more suitable for such multimodal concepts. Also, the tolerability was also very good with no substantial increase in toxicity from LAT.
Comments on the Quality of English Language
Please see above
Author Response
Comment1: Optionally, the authors could consider to also mention the CURB study (NCT03808662), which was the first randomized study to demonstrate benefit from LAT in oligoprogressive NSCLC including tumors with oncogenic drivers. Of note, no benefit was noted for breast cancer, which shows that lung cancer maybe more suitable for such multimodal concepts. Also, the tolerability was also very good with no substantial increase in toxicity from LAT.
Thank you very much for your important and constructive comment. In response, we have added the following detailed description of the CURB trial to section 4.3. Current status of clinical trials for Oligo-PD:
"In 2024, the results of the CURB trial—the first randomized study globally to specifically target patients with Oligo-PD. This randomized phase II trial enrolled patients with breast cancer or NSCLC presenting with 1–5 progressive lesions (Oligo-PD) and compared the efficacy of LAT to all progressive lesions (LAT arm) versus physician’s choice of systemic chemotherapy (chemotherapy arm). The primary endpoint, PFS, was significantly prolonged in the LAT arm compared to the chemotherapy arm, with median PFS of 7.2 months versus 3.2 months, respectively (HR 0.53, P = 0.0035). Notably, in the subgroup analysis of patients with NSCLC, the median PFS was 10.0 months in the LAT arm and 2.2 months in the chemotherapy arm, again demonstrating a significant benefit in favor of LAT (P = 0.001). In contrast, OS did not differ significantly between the treatment arms in the overall population or within the NSCLC and breast cancer cohorts. This lack of OS difference has been attributed to several factors, including the relatively short follow-up period and substantial heterogeneity in baseline assessment methods, treatment lines, therapeutic modalities, and tumor burden62.